# Numerical Investigation of Flow and Heat Transfer over a Shallow Cavity: Effect of Cavity Height Ratio

**Salem S. Abdel Aziz [1,2,*]** and **Abdel-Halim Saber Salem Said [2,3]**

[1] Mechanical Engineering Department, King Abdulaziz University, Jeddah 21589, Saudi Arabia
[2] Mechanical Power Engineering Department, Faculty of Engineering, Zagazig University, Zagazig 44519, Sharkia, Egypt; asssaed@uj.edu.sa
[3] Department of Mechanical and Material Engineering, University of Jeddah, Jeddah 21589, Saudi Arabia
[*] Correspondence: salemanos@kau.edu.sa; Tel.: +966-01-538-582-392; Fax: +966-01-467-6652

**Abstract:** Flow over shallow cavities is used to model the flow field and heat transfer in a solar collector and a variety of engineering applications. Many studies have been conducted to demonstrate the effect of cavity aspect ratio (AR), but very few studies have been carried out to investigate the effect of cavity height ratio (HR) on shallow cavity flow behavior. In this paper, flow field structure and heat transfer within the 3-D shallow cavity are obtained numerically for two height ratio categories: HR = 0.0, 0.25, 0.5, 0.75, and 1.0 and HR = 1.25, 1.5, 1.75, 2.0, 2.25, and 2.5. The governing equations, continuity, momentum, and energy are solved numerically and using the standard (K-ε) turbulence model. ANSYS FLUENT 14 CFD code is used to perform the numerical simulation based on the finite volume method. In this study, the cavity aspect ratio, AR = 5.0, and Reynolds number, Re = $3 \times 10^5$, parameters are fixed. The cavity's bottom wall is heated with a constant and uniform heat flux (q = 740 W/m$^2$), while the other walls are assumed to be adiabatic. For the current Reynolds number and cavity geometry, a single vortex structure (recirculation region) is formed and occupies most of the cavity volume. The shape and location of the vortex differ according to the height ratio. A reverse velocity profile across the recirculation region near the cavity's bottom wall is shown at all cavity height ratios. Streamlines and temperature contours on the plane of symmetry and cavity bottom wall are displayed. Local static pressure coefficient and Nusselt number profiles are obtained along the cavity's bottom wall, and the average Nusselt number for various height ratios is established. The cavity height ratio (HR) is an important geometry parameter in shallow cavities, and it plays a significant role in the cavity flow behavior and heat transfer characteristics. The results indicate interesting flow dynamics based on height ratio (HR), which includes a minimal value in average Nusselt number for HR ≈ 1.75 and spatial transitions in local Nusselt number distribution along the bottom wall for different HRs.

**Keywords:** numerical; forced convection; cavity height ratio; flow structure

## 1. Introduction

Flow over cavities is used to model the flow field in a wide range of engineering applications and practical devices [1–4]. Solar energy collectors, electronic cooling systems, ribbed channel flows, combustion chambers, food processing, lubrication technologies, nuclear reactors, turbine-blade tip flows, and environmental issues are all examples of flow over cavities. Furthermore, cavity flow represents a very important example of separated flow, which continuously receives more interest to understand its nature. Rectangular symmetric cavities with (length (L), width (W), and front and back wall height (H)) are generally classified according to their length-to-height ratio (aspect ratio, AR = L/H) and length-to-width ratio (width ratio, WR = L/W). Independently of the nature of the flow field inside it, the cavity is said to be deep if AR < 1 and shallow if AR > 1, two-dimensional if WR < 1, and three-dimensional if WR > 1 [5]. In this study, a rectangular asymmetric cavity with dimensions (length (L), width (W), front wall height ($H_1$), and back wall height

($H_2$)) was classified based on length-to-front wall height ratio (aspect ratio, AR = $L/H_1$), length-to-width ratio (width ratio, WR = $L/W$), and back wall height-to-front wall height (height ratio, HR = $H_2/H_1$). Another classification with more physical insight is related to the location of the reattachment point of the shear layer. The cavity is said to be open when the reattachment takes place near the downstream corner and closed when the reattachment point is located on the floor of the cavity. For subsonic flows, cavities are found to be open for AR < 6–8 and closed for AR ≥ 9–15 [6].

Several experimental studies on rectangular open symmetric cavities (i.e., with the same heights, $H_1 = H_2$) have been conducted to better understand the flow structure and heat transfer characteristics inside cavities. Yamamoto et al. [7] measured the flow pattern and heat transfer characteristics of laminar and turbulent flow on the cavity's bottom surface at different aspect ratios. A correlation between mean Nusselt number, Reynolds number, and aspect ratios was obtained for laminar to turbulent heat transfer regions. Sinha et al. [8] conducted experiments to investigate the results of laminar flows over cavities (deep and shallow). They demonstrated that the aspect ratio influenced the vortices' shapes and numbers within the cavity. Aung [9] considered experimental laminar forced convection in cavities (AR = 1.0 and 4.0), but fluid dynamical structures received little attention. It was found that the local heat transfer had a maximum value on the cavity bottom wall between the midpoint and downstream wall. Richards et al. [10] reported heat transfer for turbulent flow within 2-D cavities with a low aspect ratio heated from the bottom. The experimental results showed that the cavity aspect ratio had a large effect on heat transfer, but the thickness of the inlet boundary layer thickness at the separation point had a smaller effect. Metzger et al. [11] tested the turbulent flow with convection heat transfer on cavities in a narrow flow channel experimentally. They demonstrated that the flow pattern was strongly influenced by the aspect ratio and had little influence from Reynolds number changes. Esteve et al. [12] tested the turbulent flow at low Reynolds number over a rectangular cavity with aspect ratio AR = 10. The stagnation region and the flow evolution of the cavity downstream and upstream were the focus of the study. The results show that the region upstream of the stagnation zone, including the separated shear layer, is unaffected by the rearward-facing step and, at the same time, by the second recirculation zone. Ozsoy et al. [13] tested laminar flow with various Reynolds numbers (Re = 4000, 9000, and 13,000) in a rectangular 2-D shallow cavity with an aspect ratio AR = 4.0. As Re increased, the downstream vortex grew larger, and the center of the vortex moved closer towards the leading edge. Experimentally, Faure et al. [14] investigated laminar flow with Reynolds numbers (Re = 1150 to 10,670) in a cavity with an aspect ratio (AR) ranging from 0.5 to 2.0 by 0.5 steps. Flow dynamical structure with vortices was created within the cavity. The results showed that the flow injection into the cavity was a two-dimensional phenomenon; however, as upstream velocity increased, flow behavior became three-dimensional. D'yachenko et al. [15,16] investigated convective heat transfer in a cavity with inclined sidewalls ($30° \leq \phi \leq 90°$) and a low aspect ratio. On the three heated cavity walls, temperature distributions were measured in longitudinal and transverse sections. For inclination ($\phi = 60°$) there was a slight increase in the mean heat-transfer coefficient averaged over the entire heated surface. Avelar et al. [17] investigated, experimentally, the flow over a shallow cavity with an aspect ratio AR = 6. The results showed that no bubble could be seen within the cavity at very low Reynolds numbers because the flow outside the cavity lacked sufficient energy. Avelar et al. [18,19] investigated the flow structure past a cavity with various parameters, such as cavity shape, Reynolds number, cavity surface roughness (using sandpaper), cavity aspect ratio, and three-dimensional effects using PIV technology. In these studies, numerical results obtained by Zdanski et al. [20] were confirmed, and good agreement between experimental and numerical results was found. Mesalhy et al. [21] investigated experimentally and numerically the flow and heat transfer over a 3-D shallow cavity with an aspect ratio from 4.0 to 17.4 at various Re. The results revealed that up to aspect ratio AR = 7.0, a single vortex was observed, whereas for aspect ratios greater than 7,

two eddies formed, one near the upstream cavity wall and the other near the downstream cavity wall. Furthermore, the flow structure influenced the local Nusselt number, and the average Nusselt number increased as the aspect ratio increased. Crook et al. [22] reported experimental results for 3-D incompressible flow over a shallow rectangular cavity with an aspect ratio AR = 6.0. The experimental results showed that the three-dimensional cavity flow was the "open-type" with an overall flow structure inside the cavity. Yang et al. [23] measured wall static pressure and sound pressure for a cavity in a low speed and low-turbulence wind tunnel. The effect of the cavity aspect ratio (AR = 1.0 to 12.0) and the width to height ratio (W/H = 1.0 and 2.0) on cavity flow structure and noise characteristics was investigated. Results showed that a cavity with aspect ratios ranging from 1 to 12 covered open and close flow patterns. Singh et al. [24] investigated experimentally and numerically turbulent flow with different Reynolds numbers over open cavities (AR = 1.0, 2.0, and 4.0) in a water channel. The recirculating zone inside the cavity interacted with downstream cavity flow for AR = 1.0, while the mass exchange between the cavity and upstream flow was observed for AR = 2.0 and 4.0.

Many numerical studies for flow and heat transfer over a rectangular symmetric cavity ($H_1 = H_2$) have been conducted. Bhatti and Aung [25] studied numerically forced convection heat transfer of 2-D laminar flow over rectangular cavities. Results indicated that the average Nusselt number changed as the aspect ratio of the cavity and the Reynolds number changed. Ooi et al. [26,27] investigated numerically the flow and heat transfer in a 2-D grooved turbine blade and 3-D ribbed ducts. Heat transfer predictions for the 3-D ribbed duct were found to be in poor agreement with experimental results. They related this to the existence of strong secondary flow structures, which turbulence models based on eddy viscosity might not be able to simulate adequately. Matos et al. [28] used a 2-D large-eddy simulation (LES) to simulate turbulent flow in a plane-symmetric cavity and over a free cavity. The data for the free cavity (AR = 2.0) was limited to displaying the pressure distribution along the cavity's floor. He concluded that the geometry has a strong influence on the dynamic behavior of the cavity, while the Reynolds number has little influence. Zdanski et al. [20,29–31] presented a series of numerical simulations of (laminar and turbulent) flow and heat transfer over 2-D shallow cavities with large aspect ratios. In these studies, the averaged governing equations of mass, momentum, and energy were solved, and the (K-ε) turbulence model was used. Regarding the flow structure, their results showed that the flow parameters were very sensitive to the change of cavity aspect ratio, the incoming flow turbulence level, and the Reynolds number. For some ranges of these parameters, the flow results revealed that the mean external flow does not touch the cavity floor. The results showed the influence of the flow structure on heat transfer and the opposite behavior of the displacement of the two vortices inside the cavity for turbulent and laminar regimes within the cavity. Yao et al. [32] simulated the unsteady laminar flow over 3-D deep and shallow cavities numerically to examine the effects of cavity geometry, Reynolds number, and flow inlet conditions on the cavity flow field. Results showed that the flow within the cavity consists of a large recirculation zone only. No other flow structures were evident in the cavity cross-sectional plane. Alammar [33] simulated 2-D turbulent pipe flow with cavities to investigate the effect of cavity aspect ratio on the flow and heat transfer characteristics. The standard (K-ε) turbulence model was used in the numerical simulation. From the solution, there was circulation within the cavity. Heat transfer enhanced while pressure drop increased in cavities with a higher aspect ratio. Numerically, Stalio et al. [34] investigated laminar flow with heat transfer in a periodic series of shallow cavities in a channel at a low Prandtl number. A stable vortex formed downstream of the cavity's backward step, resulting in a negative effect on the heat transfer. Reynold number enhances the vortex's insulating effect. When the cavity aspect ratio, AR = 10 and Pr = 0.71, the global Nusselt number increases with the Reynolds number. The influence of incoming flow conditions (wall jet inflow and boundary layer flow) on turbulent flow over a shallow cavity were studied numerically by Arous et al. [35]. The low-Re stress-omega model was used in the numerical approach. Within the cavity,

the simulation revealed three eddy recirculation zones. The lengths of reattachment were shorter in the wall jet upstream case than in the boundary layer flow case. Arous et al. [36] investigated the effect of cavity depth (H) on flow structure over a cavity with aspect ratio (AR = 10) for a 2-D turbulent wall jet. The results show that the flow structure is very sensitive to the cavity depth to nozzle height ratio (H/b); this ratio causes a decrease in reattachment length. Numerically, Maheandera and Padmanaban [37] simulated 2-D laminar wall jet flow with low Reynolds numbers in a shallow cavity with different aspect ratios (AR) and upstream step lengths. The vortex structure shifted from the leading edge to the trailing edge of the wall as Re increased. As the step length was decreased, the distance between vortices was reduced. When Re increased, the maximum temperature contour distributions in shallow cavity regions and the highest convection heat transfer were achieved in heated walls. Arous [38] considered two configurations of the incoming flow, a boundary layer flow, and a plane wall jet flow, to examine heat transfer over a shallow cavity. A heat transfer enhancement was observed in the wall jet incoming flow event in comparison to a boundary layer. Likewise, it was found that increasing the cavity depth to the jet nozzle height ratio improved, even more, the heat transfer. The maximum heat transfer occurred upstream of the reattachment. In Arous [39], the effect of the cavity aspect ratio (AR from 1.0 to 14) on flow and heat transfer characteristics was investigated numerically at two different Reynolds numbers. As the aspect ratio increased, the heat transfer improved, and the flow pattern structure changed heavily. In addition, the increase of the Reynolds number did not affect the flow structure but improved the heat transfer. A correlation between the local Nusselt number and the velocity fluctuations profiles was obtained.

The rectangular asymmetrical cavity geometry (i.e., $H_2 \neq H_1$) represents the configuration for some engineering applications such as the railway equipment encountered over the roof of many types of trains and Channels equipped with different height ribs. However, to our knowledge, there have been very few studies on such geometry. Yamamoto et al. [40] conducted an experiment to investigate the effect of cavity height ratios (HR = $H_2/H_1$) ranging from zero to one on the flow and heat transfer characteristics inside a deep cavity with aspect ratio AR = 1.0. The vortex flow inside the cavity was varied according to the cavity's height ratio (HR) and Reynolds number, Re. Results showed that the flow pattern for cavity height ratio HR = 0.8 was completely changed at Re = $1.5 \times 10^4$. Cornu et al. [41] investigated the effect of cavity height ratio (HR = 0.8925, 1.1075) on flow features in 2-D deep cavities of aspect ratio AR = 0.2419 and 0.6452 using wall pressure and PIV measurements. When the cavity height ratio was changed, the results showed that it had a significant effect on the cavity flow structure and the pressure at the wall.

Based on the above comprehensive review, the majority of the previous studies have been carried out for the rectangular symmetric cavities (i.e., the front wall height $H_1$ and back wall height $H_2$ are equal) with minimum attention to asymmetrical cavity geometry ($H_2 \neq H_1$). Thus, the motivation of the present work is to improve the knowledge about the asymmetric shallow cavity flow features. The objectives are fulfilled by investigating the effect of cavity height ratio (HR) on the flow field structure and heat transfer characteristics within a 3-D shallow cavity with a fixed aspect ratio (AR = 5.0). In this study, a shallow cavity with two categories of height ratios was considered: HR = 0.0, 0.25, 0.75, and 1.0 and HR = 1.25, 1.5, 1.75, 2.0, 2.25, and 2.5. A numerical simulation with a two equation standard (K-ε) turbulence model was used to solve the governing Naiver–Stokes equations and energy equation for steady, incompressible flow. The flow field streamline and temperature contours inside the cavity could be displayed. Furthermore, the local static pressure coefficient and Nusselt number along the cavity bottom wall could be calculated.

## 2. Mathematical Formulations

### 2.1. Problem Geometry

The problem considered in this study is a 3-D open shallow cavity located at the bottom of a channel, as shown in Figure 1. The geometry consists of three sections: upstream channel, cavity, and downstream channel. The length is $L_i$ = 40 cm for the upstream channel, and the vertical height is $H_i$ = 40 cm. For the cavity section, the front wall height is $H_1$ = 8 cm; the bottom wall length is L = 40 cm, and the back wall height, $H_2$, is changed from zero to 20 cm, with steps of 2 cm. For the downstream channel, the length is $L_e$ = 100 cm, and the vertical height is $H_e$, which changes according to the change of the cavity back wall height, $H_2$. Thus, the span-wise width of the geometry is W = 40 cm. The cavity bottom wall is made up of a 3 mm thick aluminum plate and is maintained at uniform constant heat flux (q = 740 W/m$^2$). The remaining walls of the channel and cavity are assumed adiabatic.

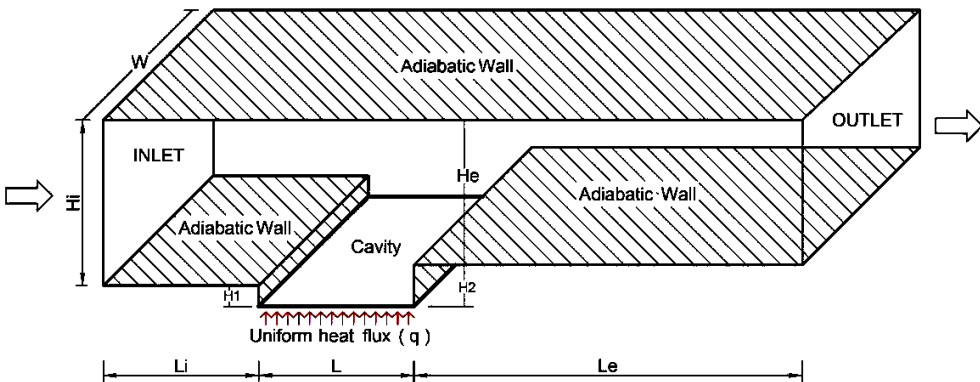

**Figure 1.** Schematic diagram with dimensions for the shallow cavity in a channel.

### 2.2. Governing Equations and Turbulence Modeling

The numerical simulation of a steady, incompressible turbulent flow over three-dimensional shallow cavity is considered. The governing equations for such a flow are the time average of the continuity, the Navier–Stokes equation for momentum and the energy equation for heat transfer, which can be written as follows:

The continuity equation is

$$\frac{\partial u_i}{\partial x_i} = 0 \tag{1}$$

where $u_i$ is the velocity component in the i direction.

The momentum equation is

$$\frac{\partial \left( u_i u_j \right)}{\partial x_j} = -\frac{1}{\rho} \frac{\partial p}{\partial x_i} + \frac{\partial}{\partial x_j} \left[ (\nu + \nu_t) \frac{\partial u_i}{\partial x_j} \right] \tag{2}$$

where, $\rho$ is the fluid density, p is the pressure, $\nu$ is the kinematic viscosity, and $\nu_t$ is the turbulent kinematic viscosity.

The energy equation is

$$\frac{\partial (u_i T)}{\partial x_i} = \frac{\partial}{\partial x_j} \left[ \left( \frac{\nu}{Pr} + \frac{\nu_t}{\sigma_t} \right) \frac{\partial T}{\partial x_j} \right] \tag{3}$$

where T is the fluid temperature, Pr is the Prandtl number, and $\sigma_t$ is the turbulent Prandtl number.

The standard (K-ε) turbulence model is one of the most common turbulence models used in different applications [42]. The equations of turbulence kinetic energy, K, and rate of dissipation of turbulence energy, ε, are as follows:

$$\frac{\partial(u_i K)}{\partial x_i} = \frac{\partial}{\partial x_j}\left[\left(\nu + \frac{\nu_t}{\sigma_k}\right)\frac{\partial K}{\partial x_j}\right] + \frac{G_k}{\rho} - \varepsilon \tag{4}$$

$$\frac{\partial(u_i \varepsilon)}{\partial x_i} = \frac{\partial}{\partial x_j}\left[\left(\nu + \frac{\nu_t}{\sigma_\varepsilon}\right)\frac{\partial \varepsilon}{\partial x_j}\right] + C_{1\varepsilon}\frac{\varepsilon G_k}{\rho K} - C_{2\varepsilon}\frac{\varepsilon^2}{K} \tag{5}$$

where $G_k$ is the production of turbulent kinetic energy due to the mean velocity gradient. $G_k$ is given by

$$G_k = -\overline{\rho \acute{u}_i \acute{u}_j}\left(\partial u_i / \partial x_j\right) \tag{6}$$

The expression $(-\overline{\rho \acute{u}_i \acute{u}_j})$ is the Reynolds shear stress. The turbulence viscosity $\nu_t$ is given as

$$\nu_t = C_\mu\left(\frac{K^2}{\varepsilon}\right) \tag{7}$$

The model constants are $\sigma_k = 1.0$, $\sigma_\varepsilon = 1.3$, $C_{1\varepsilon} = 1.44$, $C_{2\varepsilon} = 1.92$, and $C_\mu = 0.09$. The standard wall function is used in conjunction with the (K-ε) model to bridge the viscosity-affected region beside the solid walls.

### 2.3. Boundary Conditions

Due to geometric symmetry with respect to the longitudinal-normal center plane, only half of the geometry is considered in the numerical solution. The computational domain and boundary conditions are shown in Figure 2. On the plane of symmetry (z = 0), the symmetry boundary condition is applied. The airflow direction into the channel is from left to right at constant temperature $T_{in} = 300$ K. The inlet velocity profile shown in Figure 3 is obtained from the outlet velocity of a separate numerical solution of airflow in a square channel with a uniform inlet velocity of $U_{in} = 12$ m/s. The channel has the same cross-sectional area (40 cm × 40 cm) as the current simulation and is long enough to achieve a fully developed flow at the exit plane. The inlet flow turbulence intensity is specified as 10% of the mean flow kinetic energy. The exit boundary condition is considered a pressure outlet boundary, where the pressure is specified, and all the other flow properties are extrapolated from the internal cells. A no-slip boundary condition is applied at all walls. A constant heat flux boundary condition is applied at the cavity bottom aluminum wall, and the heat diffusion within the aluminum plate is considered. All other walls are considered adiabatic.

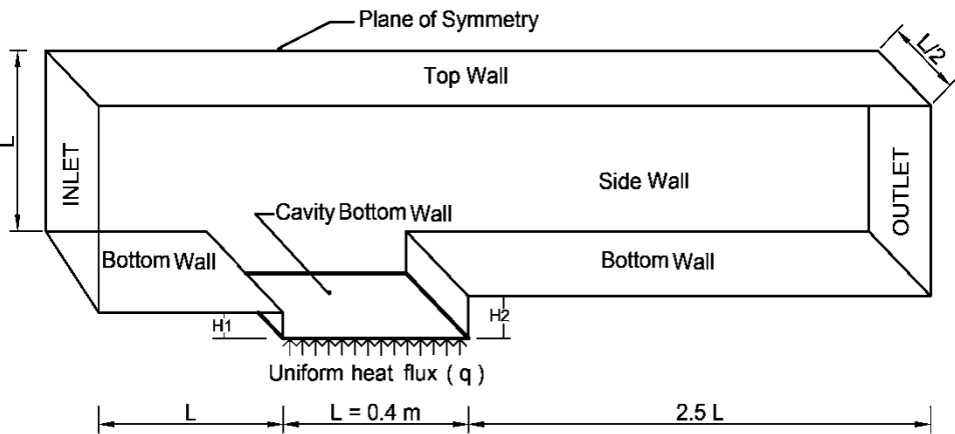

**Figure 2.** Computational domain and boundary conditions.

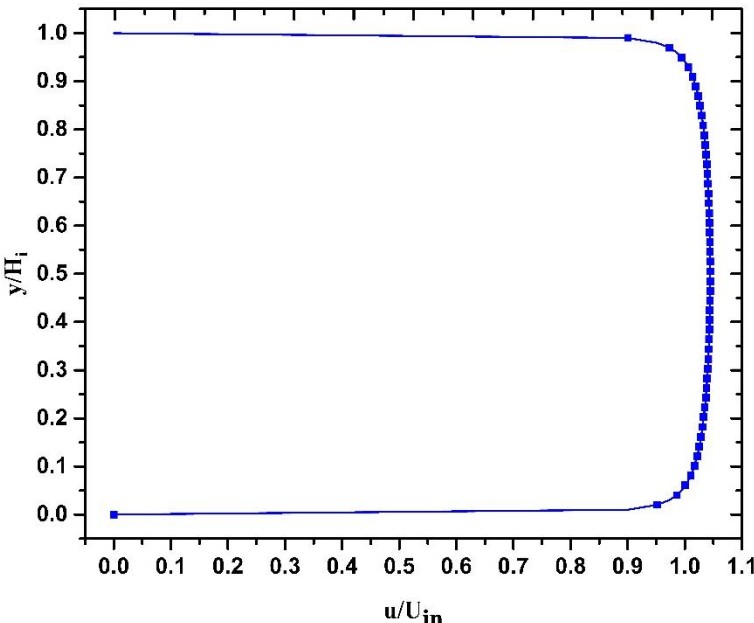

**Figure 3.** Fully developed inflow velocity profile at the vertical symmetry plane.

### 2.4. Numerical Procedures

In this study, the numerical simulations are performed to obtain the heat transfer and fluid flow over a shallow cavity for steady-state operating conditions. The numerical simulation is based on solving Reynold's averaged Navier–Stokes equation on a 3-D geometry. The governing equations are solved for turbulent flow and heat transfer. The airflow is assumed to be incompressible since the maximum flow velocity is around 12 m/s, which corresponds to a low-speed subsonic flow (Ma = 0.034), and the temperature change is within 20 K. Other air physical properties are assumed constant and evaluated for air at the inlet temperature of $T_{in}$ = 300 K ($\rho$ = 1.225 kg/m$^3$, $C_p$ = 1006.43 J/(kg·K), $\mu$ = 1.7894 × 10$^{-5}$ kg/(m·s), and k = 0.0242 W/(m·K)). The viscous dissipation, the radiation heat transfer, and the body forces—therefore the natural convection—are neglected. CFD software, the FLUENT commercial code version 14, was used to solve the set of governing equations, and the standard (K-$\varepsilon$) turbulence model with the wall function was selected in the simulations. The pressure–velocity coupling SIMPLE (semi-implicit method for pressure-linked equations) algorithm is used in the solution, and a second-order accurate finite difference scheme is used in the discretization of the pressure correction equation. In contrast, for momentum, energy, and turbulence equations, a second-order upwind scheme is used. In the numerical simulation, two criteria satisfy the convergence: the first is based on the normalized residual of each variable is less than 10$^{-8}$, and the second is based on the average temperature of the heated cavity bottom wall, being almost constant. The results are obtained once the solution has been converged.

### 2.5. Flow and Heat Transfer Relations

The Reynolds number is calculated based on the inlet air velocity $U_{in}$ and the cavity bottom wall length L as

$$\text{Re} = \frac{U_{in}L}{\nu} \tag{8}$$

where $\nu$ is the kinematic viscosity of the fluid.

The static-pressure coefficient $C_p$ is calculated along the centerline of cavity bottom wall as

$$C_p = \frac{p - p_o}{\left(\frac{1}{2}\right)\rho U_{in}^2} \tag{9}$$

where p is the static wall pressure and $p_o$ is the reference pressure extracted from the free stream flow.

The local Nusselt number $Nu_x$ on the heated cavity bottom wall can be defined as

$$Nu_x = \frac{hL}{k} \tag{10}$$

where k is the thermal conductivity, and h is the heat transfer coefficient, which is defined on the bottom surface of the cavity as

$$h = \frac{q}{(T_w - T_{in})} \tag{11}$$

where q is the wall heat flux, $T_w$ is the local wall temperature, and $T_{in}$ is the inlet flow temperature.

The average Nusselt number $Nu_{ave}$ is calculated by integrating the local Nusselt number along the bottom wall of the cavity:

$$Nu_{ave} = \frac{1}{L} \int_0^L Nu_x dx \tag{12}$$

### 2.6. Grid Structure and Code Validation

The computational grid shown in Figure 4 was created in ANSYS Mesh. Structured quadrilateral cells were generated at all domains. First, a boundary layer mesh was generated with biased growing layers in the vicinity of all walls, where the no slip boundary condition is valid. For the boundary layer mesh over the cavity bottom wall, the first layer thickness was 0.005 mm, which yielded a dimensionless wall distance (y+ < 0.5). This is a requirement to correctly resolve the viscous boundary layer with the selected turbulence model and the standard wall function [43]. Next, the mesh profiles generated for all other wall surfaces of the domain had fine mesh profiles to satisfy the y+ criteria. The quality of the mesh is reduced at locations where strong gradients in velocity fields and temperature do not exist.

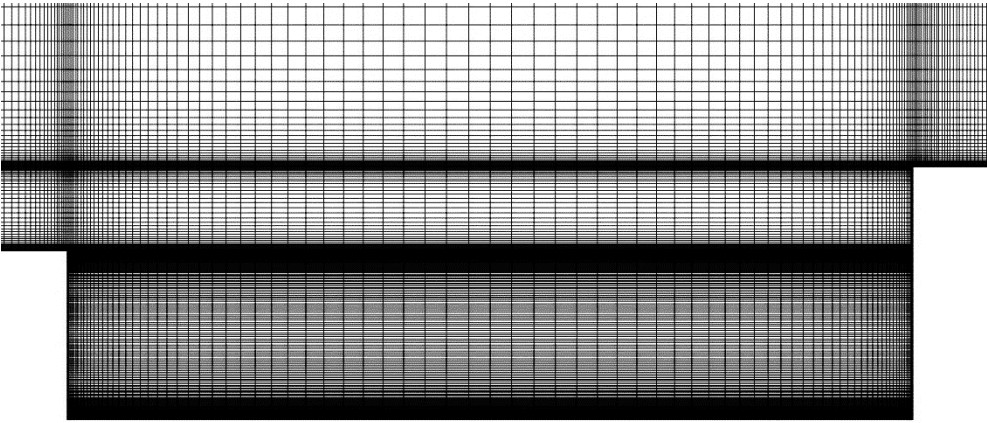

**Figure 4.** Mesh distribution of the computational domain.

The sizing parameters, as mentioned earlier, were determined after a systematic grid independence study to check the grid independence of the solutions. Three different grid sizes (128,289), (275,380), and (1,265,301) were used and compared to a higher resolution grid. Figure 5 shows the pressure coefficient distribution along the cavity bottom wall of the rectangular symmetric cavity with a height ratio HR = 1.0 for different grid sizes compared with experimental data [21]. The experimental data were for a case of the cavity aspect ratio of AR = 5.4 compared to the aspect ratio of AR = 5.0 for the current study, whereas the Reynold's number was almost the same. The results of the $C_p$ profile showed

reasonable agreement with experimental data with a very similar trend. The averaged Nusselt numbers along the cavity bottom wall were 860.6, 867.4, and 870.7 for the three grid sizes used. The final grid utilized did not yield a change of more than 0.25% for the average Nusselt number. For all cases in the present study, the mesh count was approximately $(1.3 \times 10^6)$ for the solution domain.

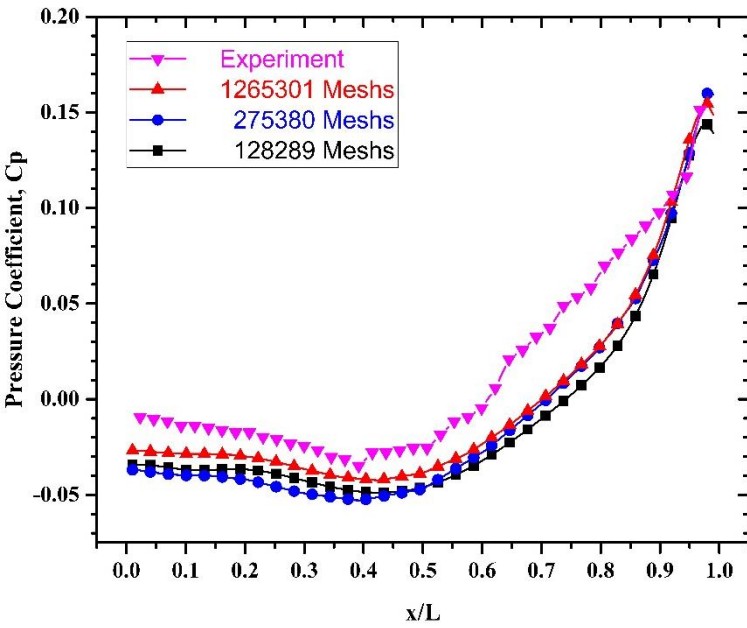

**Figure 5.** Comparison of pressure coefficient with experiment result of Mesalhy et al. [21].

### 3. Results and Discussion

Numerical simulations using (K-ε) turbulence were performed for 3-D turbulent flow and heat transfer over a shallow cavity heated from the bottom at a constant heat flux. The cavity aspect ratio (AR = 5.0) and Reynolds number (Re = $3 \times 10^5$) parameters were fixed. The results for two categories of cavity height ratios (HR = 0.0, 0.25, 0.5, 0.75, and 1.0 and HR = 1.25, 1.5, 1.75, 2.0, 2.25, and 2.5) were investigated. Flow field structures (streamlines and velocity profiles) and temperature contours are presented within the cavity. The static pressure coefficient, local Nusselt number, and average Nusselt number were obtained at the cavity bottom wall for different cavity height ratios.

### 3.1. Flow Fields Results

Figure 6 shows the flow field streamlines inside the cavity on the plane of symmetry at different cavity height ratios (0.0 ≤ HR ≤ 1.0). For the current study with cavity aspect ratio (AR = 5.0) and turbulent Reynolds number, there was no flow impingement on the cavity bottom wall. Due to stronger outside flow, only one large recirculating vortex formed inside the cavity and rotated in the reversed direction. For the asymmetric cavities depending on the height ratio (HR < 1.0), the vortex shape differed from the symmetric cavity case of (HR = 1.0). In general, the vortex shape was oval and had a long horizontal dimension.

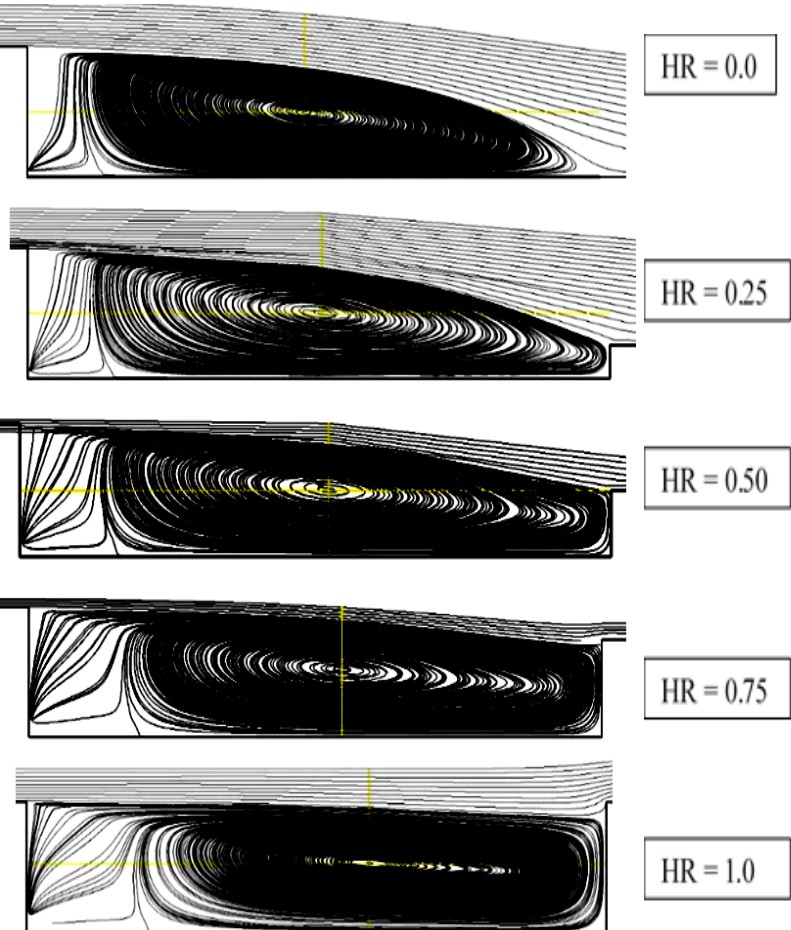

**Figure 6.** Streamlines inside the shallow cavity with (AR = 5.0) at different height ratios (0.0 ≤ HR ≤ 1.0).

Figure 7 shows the flow field streamlines inside the cavity on the plane of symmetry at various cavity height ratios (1.25 ≤ HR ≤ 2.5). Higher height ratios caused more flow momentum to entrain into the cavity from the back towards the front. At values of (HR > 1.75), the longitudinal length of the vortex decreased, and the vortex form became more confined. The positions of the vortex center at different height ratios are depicted in Figure 8. The results showed that the center of the recirculation zone was located between the mid and cavity back wall. It was noticed that the vortex center shifted slightly to the right towards the back wall and slightly vertically upward as the cavity height ratio increased up to (HR = 1.0). For height ratios (1.25 ≤ HR ≤ 2.5) due to stronger reversed flow, the center of the vortex moved more toward the cavity back wall and shifted vertically upward.

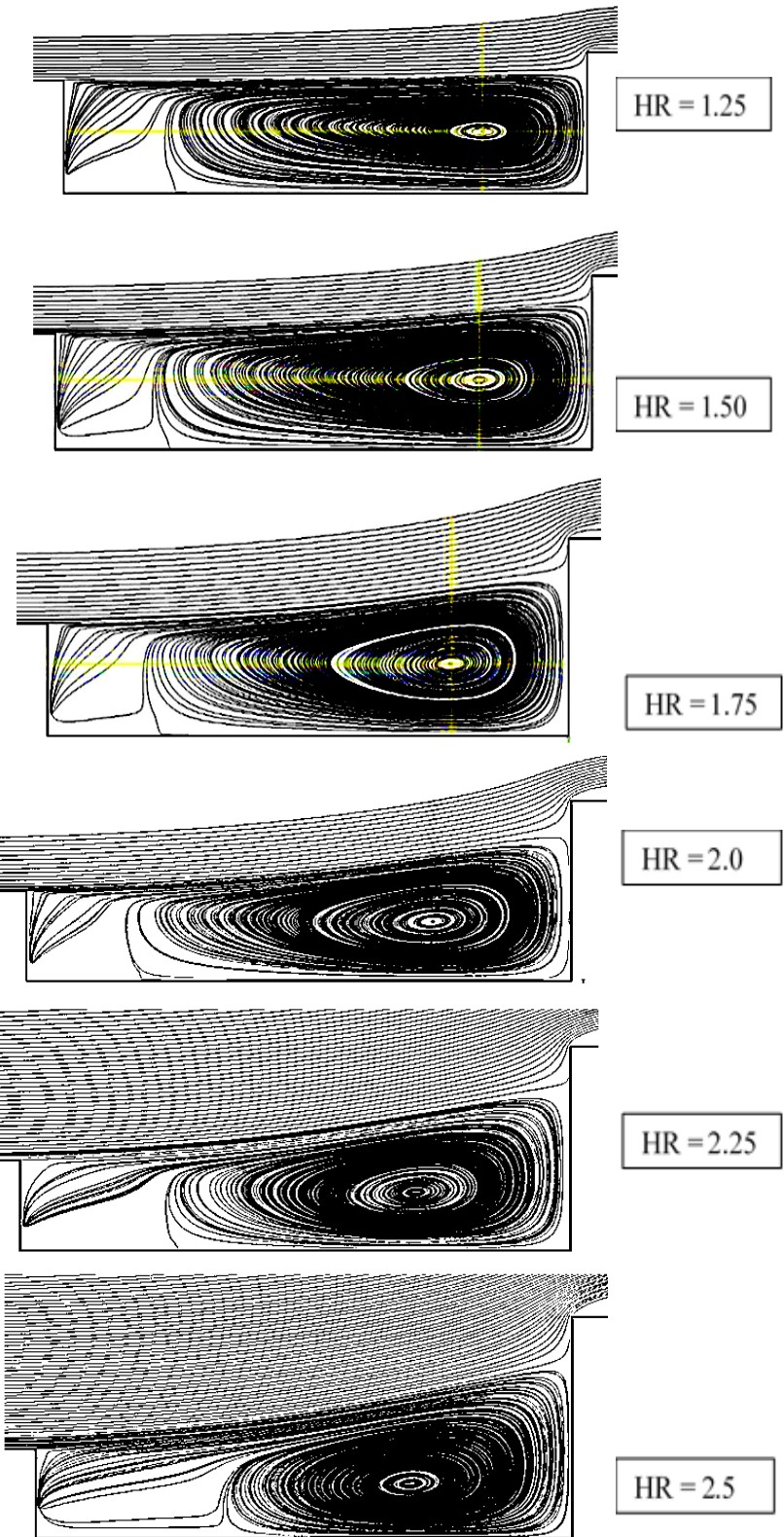

**Figure 7.** Streamlines inside the shallow cavity with (AR = 5) at different height ratios (1.25 ≤ HR ≤ 2.5).

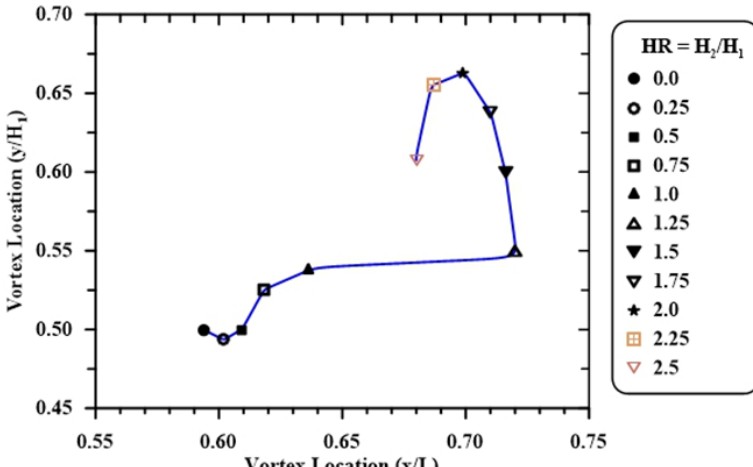

**Figure 8.** Locations of vortex center for the shallow cavity (AR = 5) at different height ratios ($0.0 \leq HR \leq 2.5$).

Velocity profiles along a vertical line passed through the vortex center for height ratios ($0.0 \leq HR \leq 1.0$) and ($1.0 \leq HR \leq 2.5$), as shown in Figure 9a,b, respectively. The velocities were normalized by a reference inlet velocity of $U_{in}$ = 12 m/s. For cavities of height ratios of ($0.0 \leq HR \leq 1.0$), the maximum reverse velocity near the cavity bottom wall increased slightly with decreasing height ratio. This could be attributed to the increase of the suction pressure near the cavity upstream wall with decreasing height ratio. For cavities with height ratios of ($1.25 \leq HR \leq 2.5$), the maximum reverse velocity near the cavity wall increased as the cavity height ratio increased. This could be attributed to the increasing stagnation pressure and higher flow momentum near the cavity back wall. The maximum value of the reverse velocity of each case indicated the strength of the recirculation zone. For example, the maximum reverse velocity of about $u \approx 0.45\ U_{in}$ occurred at a cavity height ratio HR = 2.5.

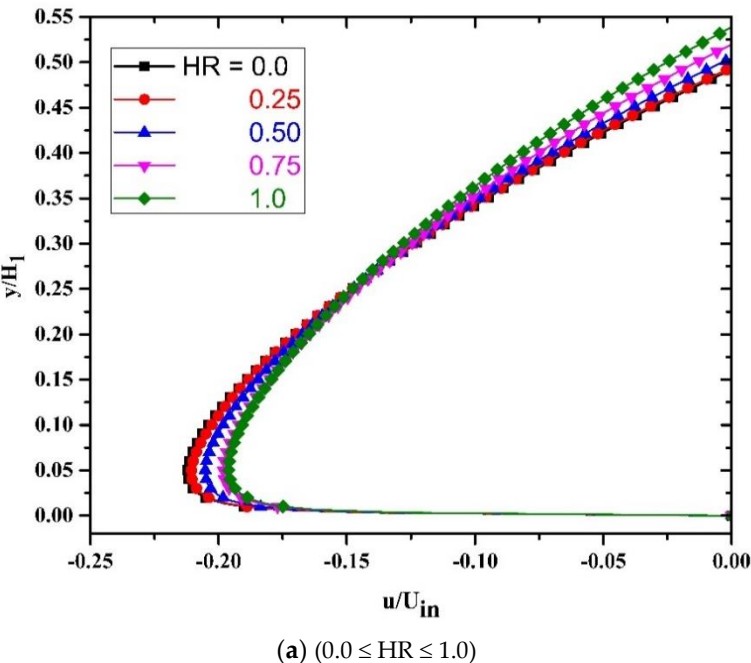

(**a**) ($0.0 \leq HR \leq 1.0$)

**Figure 9.** *Cont*.

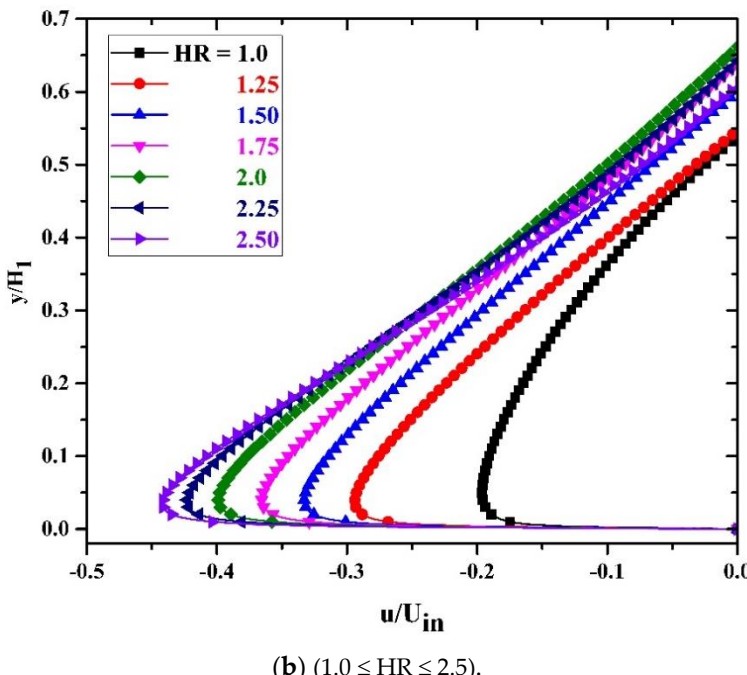

**(b)** ($1.0 \leq$ HR $\leq 2.5$).

**Figure 9.** Velocity profiles along a vertical line passes through circulation vortex center for the shallow cavity (AR = 5) at different height ratios.

Figure 10a,b display static-pressure coefficient $C_p$ distribution along the cavity bottom wall for different cavity height ratios HR. A lower-pressure region was formed behind the cavity front wall, and a higher pressure region was formed close to the cavity back wall. For height ratios ($0.0 \leq$ HR $\leq 1.0$), the downstream pressure profiles for all cases approached close values near the cavity back wall, while the upstream pressure increases with increasing the height ratio. For height ratios ($1.25 \leq$ HR $\leq 2.5$), downstream pressure and upstream pressure increased with increasing height ratio because more kinetic energy was transformed into pressure in the downstream region. The pressure difference between the two regions (upstream and downstream) at different height ratios explains the change in the strength of recirculation. The region of the lowest value of pressure coefficient was at the area underneath the center of the circulation vortex.

*3.2. Heat Transfer Results*

Figures 11 and 12 show temperature contours (isothermal lines) on the cavity bottom wall for the two categories of cavity height ratios ($0.0 \leq$ HR $\leq 1.0$) and ($1.25 \leq$ HR $\leq 2.5$), respectively. As shown in the figures, higher temperatures occurred in the region behind the front wall of the cavity due to low air velocities in these regions. For a low cavity height ratio (HR < 1.0), a good temperature distribution along the cavity bottom wall was observed as shown in Figure 11. The effect of increasing HR, as shown in Figure 12, was to increase the regions of high temperature along the cavity bottom wall. Regions of lower temperature levels occurred closer to the cavity back wall as the height ratio increased. Higher velocity gradients enhanced the convection heat transfer coefficient in the vicinity of the vortex center. Regions of lower temperature were conjugated with vortex location and size for each cavity geometry.

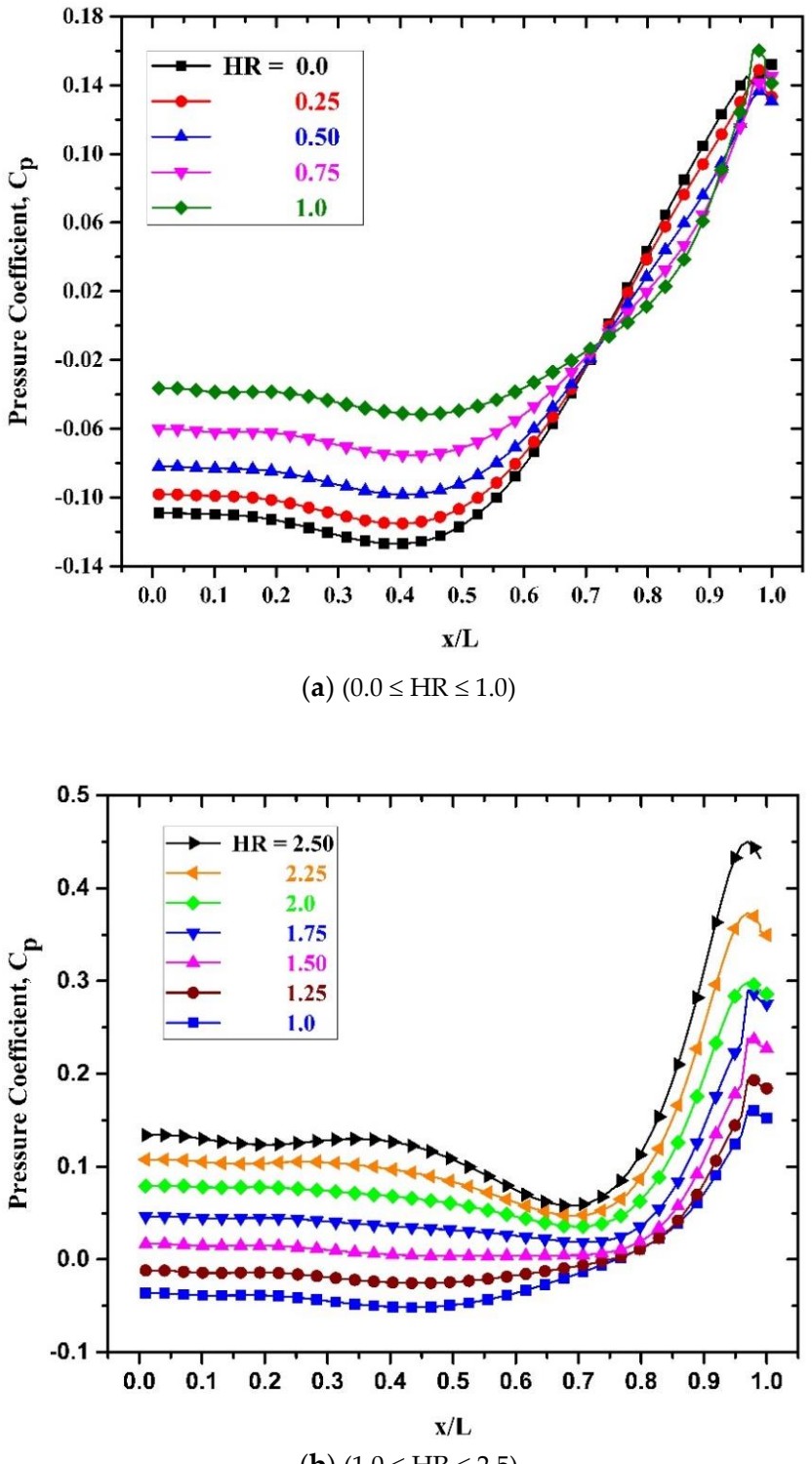

(**a**) (0.0 ≤ HR ≤ 1.0)

(**b**) (1.0 ≤ HR ≤ 2.5)

**Figure 10.** Distribution of static pressure coefficient $C_p$ along cavity bottom wall at different height ratios.

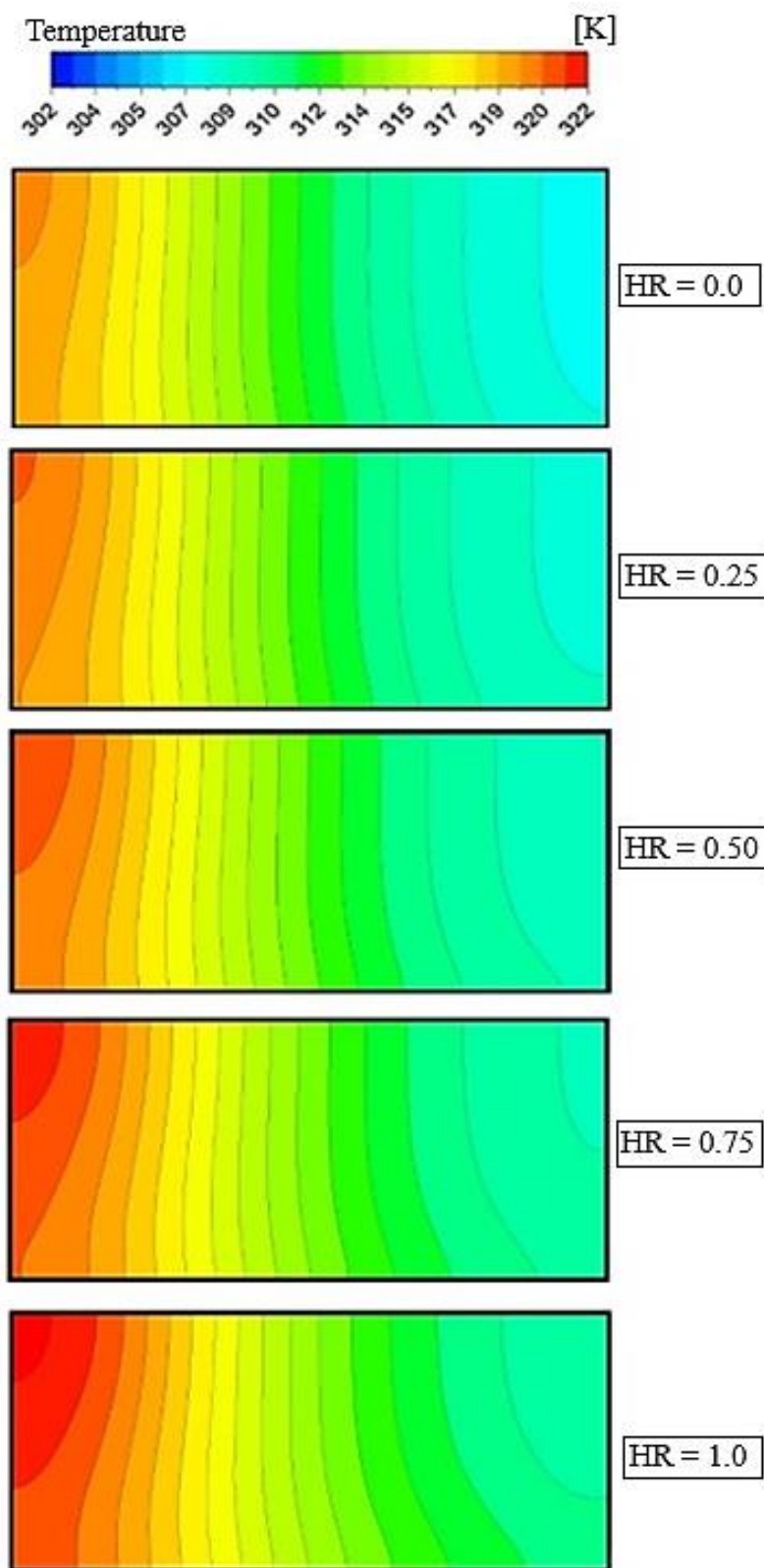

**Figure 11.** Temperature contours on the cavity bottom wall at different height ratios ($0.0 \leq$ HR $\leq 1.0$).

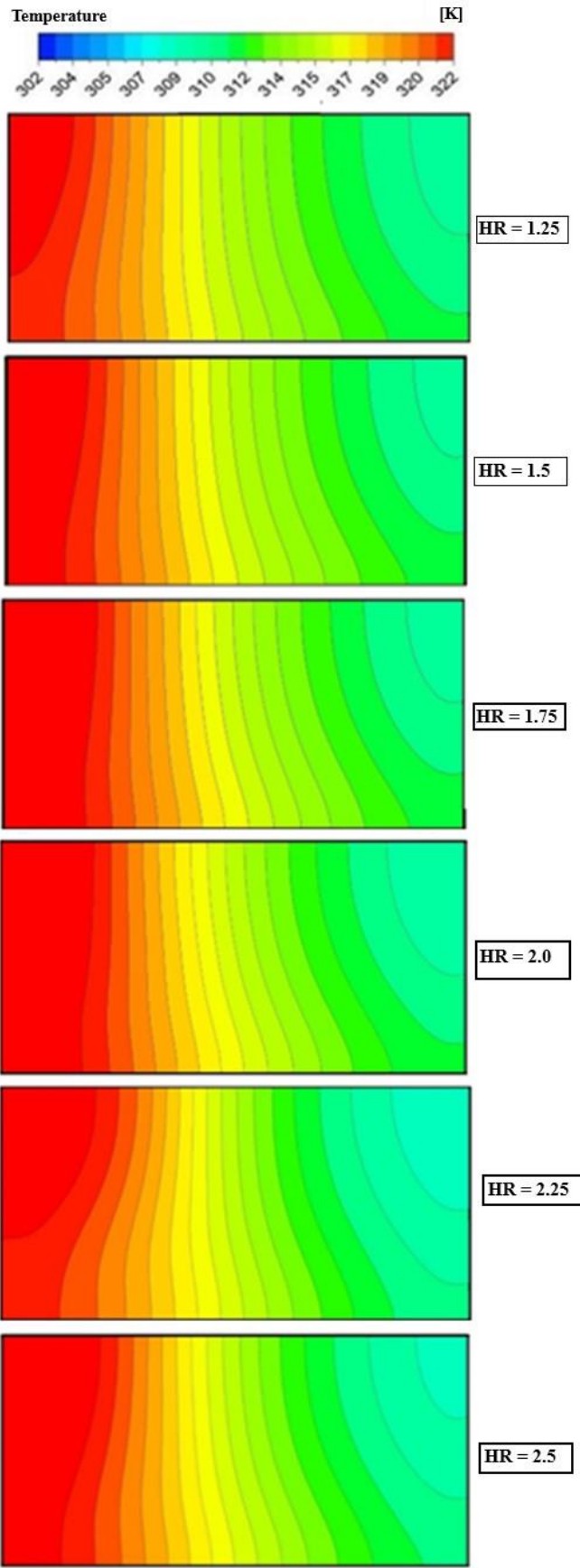

**Figure 12.** Temperature contours on the cavity bottom wall at different height ratios ($1.25 \leq HR \leq 2.5$).

Figures 13 and 14 show the temperature contours on the plane of symmetry within the cavity region at different height ratios. Higher temperature contours existed at regions near the cavity front wall where flow was almost stagnant. Lower temperature contours were formed in the downstream direction where vortex flow structures exist.

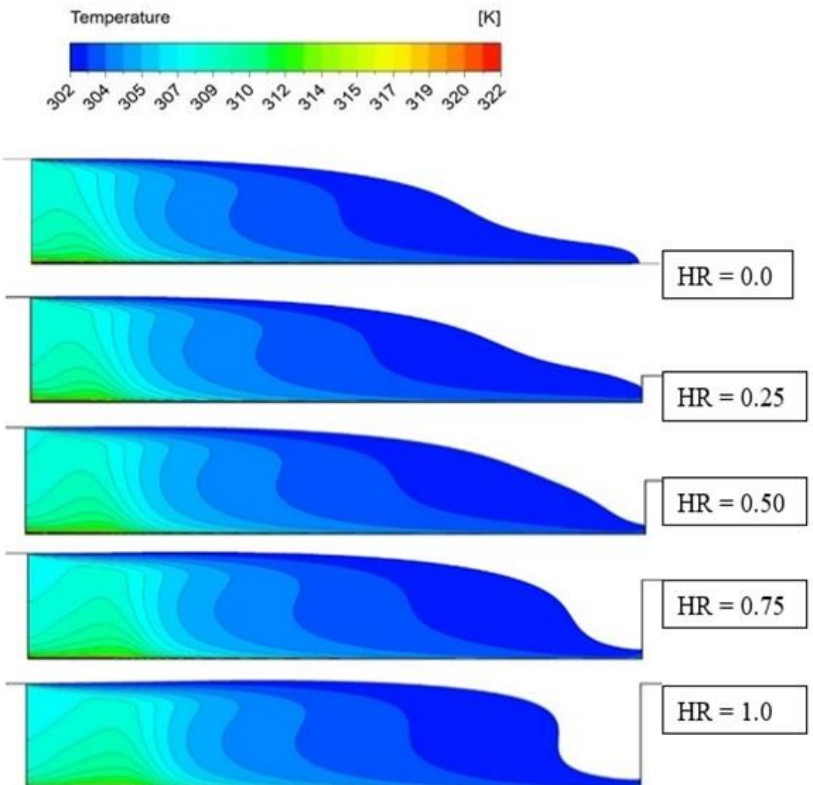

**Figure 13.** Temperature contours in a plane of symmetry for cavity at different height ratios $(0.0 \le \mathrm{HR} \le 1.0)$.

Local Nusselt number profiles, calculated at the symmetry plane on the cavity bottom wall, for different height ratios are shown in Figure 15a,b. For cavity height ratios $(0.0 \le \mathrm{HR} \le 1.0)$, Nusselt number profiles were shifted up with decreasing height ratio. This was due to the increase of the reverse velocity near the cavity bottom wall at lower height ratios, which enhanced convection heat transfer. All profiles had a similar trend, in a streamwise direction, with a lower slope (lower increase rate) in regions near front and back cavity walls. The profiles showed a higher slope (higher increase rate) over the region affected by the reverse vortex velocity. Profiles diverged moving downstream towards the cavity back wall. This indicated a stronger influence of the height ratio on the local Nusselt number closer to the cavity back wall.

For cavity height ratios $(1.25 \le \mathrm{HR} \le 2.5)$, Nusselt number profiles showed a similar trend as that of height ratios $(0.0 \le \mathrm{HR} \le 1.0)$ in the streamwise direction. However, the profiles shifted slightly down with increasing height ratios from 1.0 and reached a minimum value at HR = 1.75; this contrasted with cases of height ratios lower than 1.0. For higher height ratios HR $\ge$ 2.0, the profile started to shift above that of the height ratio of 1.75.

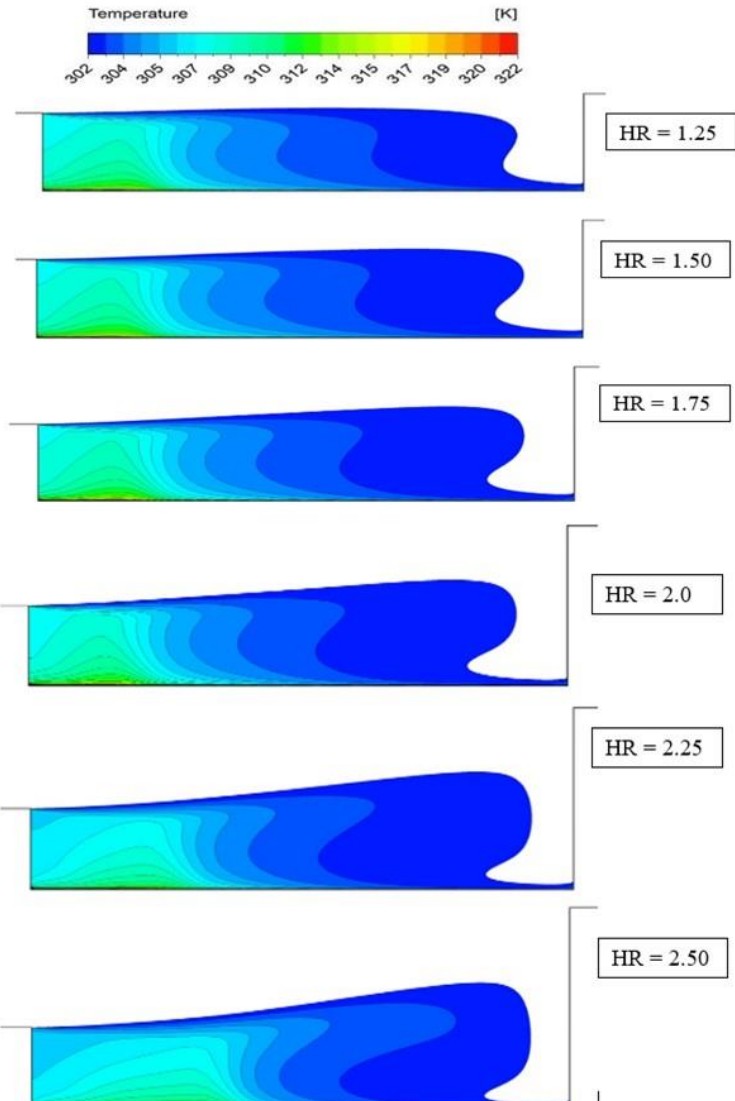

**Figure 14.** Temperature contours in a plane of symmetry for cavity at different height ratios (1.25 ≤ HR ≤ 2.5).

The average Nusselt number along bottom wall of shallow cavity (AR = 5.0) at different cavity height ratios is shown in Figure 16. The higher Nusselt number for the case of height ratio HR = 0.0 started to decrease with increasing height ratio up to a height ratio of 1.75 and then started to increase at higher height ratios of HR = 2.0, 2.25, and 2.5. The rate of change of average Nusselt number with respect to height ratio was decreasing with increasing height ratio until the inflection point at height ratio of 1.75. The decrease in average Nusselt number with increasing height ratio was due to the decrease of the swept area by the relatively cooler fluid from the free stream shear layer into the cavity, the heated wall, and the cooling deflection fluid into the recirculation zone downstream of the step. Carefully studying flow structure within the cavity can explain the change of the average Nusselt number at different cases of height ratios. For height ratios (HR) ranging from 0.0 to 1.0, a lower pressure region was formed behind the cavity front wall, which caused the flow to circulate back from cavity downstream into upstream in the form of vortex structure; this enabled relatively cooler air to entrain into the cavity as a reverse flow stream near cavity bottom wall. The formed vortex had an oval shape, with a longer dimension in the x-direction. The reverse flow stream swept a larger area of the cavity bottom wall; this enhanced the heat transfer coefficient. For height ratios HR = 1.25 up to 2.5, more cold fluid was deflected into the recirculation zone. However, the location of the vortex moved

closer to the cavity back wall. As a result, the longitudinal dimension of the vortex became shorter, and the region of stagnant flow behind the cavity front wall became larger, which in turn caused poor heat transfer over this area. For cavity height ratios HR = 2.0, 2.25, and 2.5, for the circulation zone, the reverse velocity increased due to lower pressure values and hence increased local and averaged Nusselt numbers.

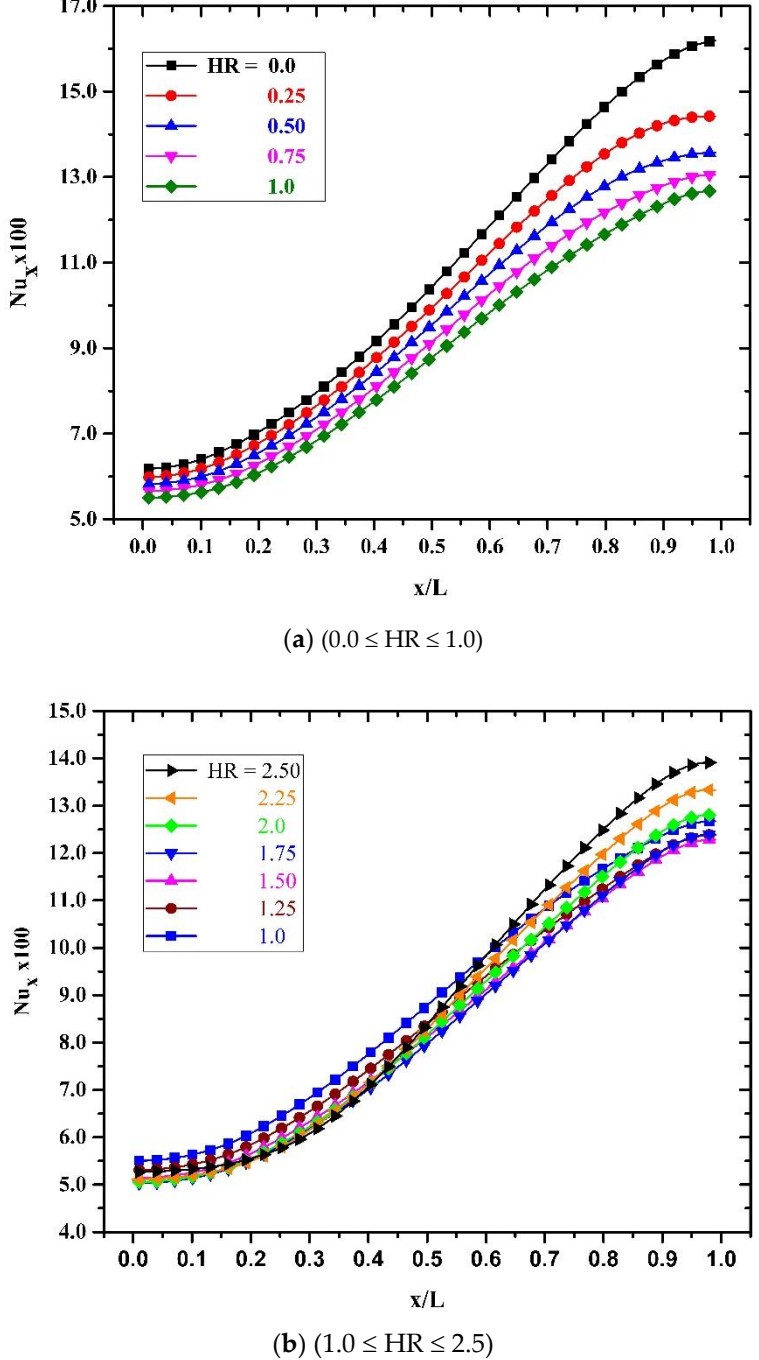

(**a**) (0.0 ≤ HR ≤ 1.0)

(**b**) (1.0 ≤ HR ≤ 2.5)

**Figure 15.** Local Nusselt number distribution along cavity bottom wall at different height ratios.

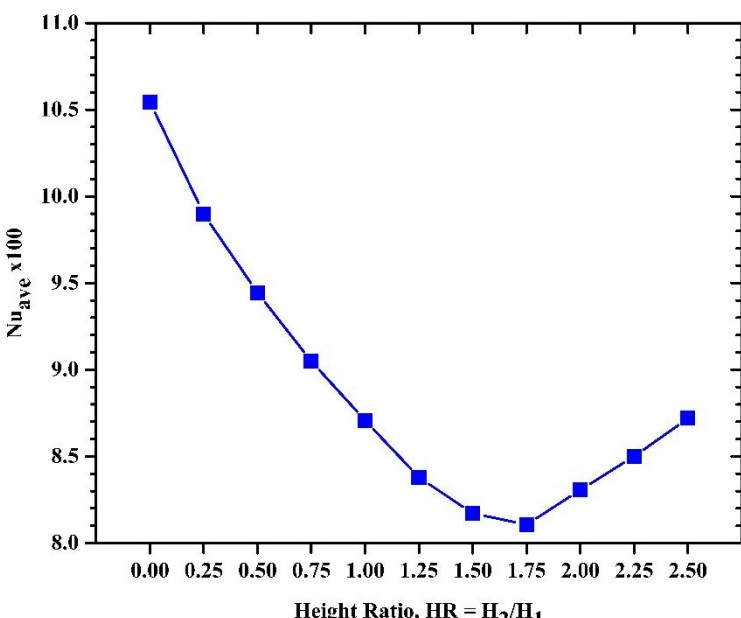

**Figure 16.** Average Nusselt number on the cavity bottom wall at different cavity height ratios.

## 4. Conclusions

Heat transfer and fluid flow characteristics in a 3-D shallow cavity with aspect ratio (AR = 5.0) at different height ratios ranging from 0.0 to 2.5 with increments of 0.25 were studied numerically. The cavity bottom wall is heated by constant heat flux, and the other walls are adiabatic. A vertical flow field structure is formed within the cavity volume. This causes the entrained air to recirculate over a wide area of the cavity's bottom wall. The shape, strength, and location of the vortex depend on the height ratio. For height ratios ($0.0 \leq$ HR $\leq 1.0$), the center locations of the vortex are approximately the same with small shifting in streamwise and normal directions. For height ratios ($1.25 \leq$ HR $\leq 2.5$), the vortex became more confined, and its center moved more closely towards the back wall of the cavity and shifted slightly up. The flow field structure explains the local pressure coefficient and local Nusselt number profiles along the cavity within the cavity. The recirculation velocity near the cavity bottom wall increases with increasing the height ratio; this enhances convection heat transfer, increasing the local Nusselt number along the recirculation zone. The average Nusselt number decreases as the cavity height ratio increases, and the minimum value of the average Nusselt number is obtained at (HR = 1.75) and then starts to increase with increasing height ratio. Based on the output results, the main conclusion of this study is that the cavity height ratio is an important geometry parameter in shallow cavities, and it plays a significant role in the cavity flow behavior and heat transfer characteristics.

**Author Contributions:** Conceptualization, S.S.A.A. and A.-H.S.S.S.; methodology, S.S.A.A. and A.-H.S.S.S.; software, S.S.A.A. and A.-H.S.S.S.; validation, S.S.A.A. and A.-H.S.S.S.; formal analysis, S.S.A.A. and A.-H.S.S.S.; investigation, S.S.A.A. and A.-H.S.S.S.; resources, S.S.A.A.; data curation, A.-H.S.S.S.; writing—original draft preparation, S.S.A.A. and A.-H.S.S.S.; writing—review and editing, S.S.A.A. and A.-H.S.S.S.; visualization, S.S.A.A. and A.-H.S.S.S.; supervision, S.S.A.A.; project administration, S.S.A.A.; funding acquisition, S.S.A.A. All authors have read and agreed to the published version of the manuscript.

**Funding:** This work was funded by the Deanship of Scientific Research (DSR), King Abdulaziz University, Jeddah, under grant No. (135-023-D1434).

**Acknowledgments:** This work was funded by the Deanship of Scientific Research (DSR), King Abdulaziz University, Jeddah, under grant No. (135-023-D1434). The authors, therefore, acknowledge with thanks DSR technical and financial support.

**Conflicts of Interest:** The authors declare no conflict of interest.

## Nomenclature

| | |
|---|---|
| AR | Cavity aspect ratio: symmetry (L/H): asymmetry ($L/H_1$), dimensionless |
| $C_p$ | Pressure coefficient along cavity bottom wall, dimensionless |
| $C_{1\varepsilon}$, $C_{2\varepsilon}$ and $C_\mu$ | Numerical constants of turbulence model |
| $c_p$ | Specific heat at constant pressure, J/(kg·K) |
| $G_k$ | Turbulence kinetic energy production |
| h | Heat transfer coefficient, W/(m$^2$·K) |
| $H_i$ | Upstream channel vertical height, m |
| $H_e$ | Downstream channel vertical height, m |
| $H_1$ | Cavity front wall height, m |
| $H_2$ | Cavity back wall height, m |
| HR | Cavity height ratio ($H_2/H_1$), dimensionless |
| i, j | Tensor notation |
| K | Turbulent kinetic energy, m$^2$/s$^2$ |
| k | Thermal conductivity, W/(m·K) |
| L | Cavity bottom wall length, m |
| $L_i$ | Upstream channel length, m |
| $L_e$ | Downstream channel length, m |
| $Nu_x$ | Local Nusselt number, dimensionless |
| $Nu_{ave}$ | Average Nusselt number, dimensionless |
| p | Static pressure along cavity bottom wall, Pa |
| Pr | Prandtl number, dimensionless |
| $p_o$ | Reference static pressure at the inlet, Pa |
| q | Heat flux rate from cavity bottom wall, W/m$^2$ |
| Re | Reynolds number based on cavity bottom length, dimensionless |
| T | Static temperature, K |
| $T_{in}$ | Inlet temperature, K |
| $T_w$ | Cavity bottom wall temperature, K |
| u | Velocity in streamwise direction, m/s |
| $u_i$ | Velocity in direction i, m/s |
| $U_{in}$ | Reference inlet velocity, m/s |
| v | Velocity normal to streamwise direction, m/s |
| W | Spanwise width of the geometry, m |
| WR | Cavity width ratio (L/W), dimensionless |
| x | Position along cavity bottom wall measured from cavity upstream wall, m |
| x, y, z | Coordinates in streamwise, cavity depth, and cavity widthdirections |

**Greek symbols**

| | |
|---|---|
| ρ | Density of fluid, (kg/m$^3$) |
| μ | Dynamic viscosity, (kg/m·s) |
| $\mu_t$ | Turbulent viscosity, (m$^2$/s) |
| ε | Rate of dissipation of turbulence kinetic energy, (m$^2$/s$^3$) |
| $\sigma_k$ and $\sigma_\varepsilon$ | Prandtl numbers for turbulence kinetic energy and rate of dissipation, respectively. |
| $\sigma_t$ | Turbulent Prandtl number |
| ν | Kinematic viscosity (m$^2$/s) |
| $\nu_t$ | Turbulent kinematic viscosity (m$^2$/s) |

**Subscripts**

| | |
|---|---|
| ave | Average |
| i | Direction, i |
| in | Channel inlet |
| t | Turbulent |
| w | Cavity heated bottom wall |
| x | Local |

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
