# Peer review of "Numerical Investigation of Flow and Heat Transfer over a Shallow Cavity: Effect of Cavity Height Ratio"

_fluids, doi:10.3390/fluids6070244_

Round 1

Reviewer 1 Report

In this study, flow field structures and heat transfer in a shallow cavity were investigated for different height-ratio categories. This is an interesting research topic as the authors explained (not extensively) the importance of this study in many applications. Here are some comments:

  1. The Introduction supports the objectives of this study; however, I suggest the authors re-organize the introduction into separate sections about the height-ratio studies, turbulence models, applications, etc.
  2. At the beginning of the Introduction, the importance of the study is not highlighted. The authors can focus more on the applications in this field and the motivation and the need to improve the knowledge in this field with such studies.
  3. Nomenclature definitely needs to be moved to a different place in the paper.
  4. The methods/equations, as well as research design, are adequately described.
  5. Figure 4, can be scaled up and more straight to clearly show the high resolution of the grid structure.
  6. The authors can add more information regarding the numerical uncertainty, convective courant number, and the selection of the turbulence model.
  7. Figure 6 can be split into two figures with additional information on the figures and in their captions.
  8. I suggest the authors focus more on the novelty of the study and their contributions to this field based on the outcomes of this study.

Author Response

First, many thanks for your response and your efforts to enrich the manuscript with your valuable comments and please open the file  the check on the response to the reviewer's comments.

Reviewer 2 Report

The manuscript presents an interesting study on the effect of front and back wall height ratios in shallow cavity, with respect to the flow shape and its effect on heat transfer. Standard numerical models from existing packages are employed. The analysis of the results is quite detailed, and supports the significance of the study.

The main drawbacks of the manuscript in its present form is that the presentation rather sloppy. The figures are rather low-quality too, although I'm not sure if this is a submission issue or an issue with the platform that lets reviewers download only a “web-ready” version of the PDF with downgraded images.

More significant improvements that need to be made to the manuscript:

Introduction:

In the first paragraphs, the uses of “were” and ”was” would be better replaced
by “have been” and “has been”. Authors should also cite at least one reference (and if possible more than one) for each of the studies that are mentioned, both for the example applications and for the numerical models employed. (Where applicable, they can be the same references discussed more in the detail in the following paragraphs of the introduction.)

2.2. Governing equations:

Please describe all quantity symbols when they are introduced, even when this
would be apparent from context (e.g. u velocity, nu viscosity,
p pressure, rho density, meaning of the overhead bar,  etc)

Equation 3: energy  equation is missing
Equation 6: I assume these would be u'_j and u'_i. please check the formulas

In the following sections there also seems to be an overlap of symbol usage,
e.g. the k for kinetic energy vs  the k for thermal dissipation

There are also several minor issues spread out throughout the manuscript. Here's a list for the first few pages:

Abstract line 20: the epsilon in k-epsilon is missing.
this issue is also present in other places in the text (e.g. line 306)

Introduction line 42, 43: check the spacing around the epsilons.

Line 62: inverted parenthesis

Line 117: extra commas. this issue is also present in several other places in the text

Line 216: is this a heading? the formatting is indistinguishable from the text

Line 227: geometry is SYMMETRIC

Lines 230-231: I think some word is missing.

Line 233: specified AS

Line 245: the numerical simulations <- remove the

Line 251: check the degree sign. also why use Celsius here where Kelvin are use dotherwise?

Line 254-256: this sentence doesn't parse correctly; a word is missing?

Author Response

First, many thanks for your response and your efforts to enrich the manuscript with your valuable comments. please find the attachment for a response to the reviewer’s comments

Reviewer 3 Report

Dear Authors,

Please see my attached comments; I enjoyed reading your manuscript. Thank you.

Author Response

(The authors gave the same response as above.)
